# CamoPatch: An Evolutionary Strategy for Generating Camouflaged Adversarial Patches

**Phoenix Neale Williams**
Department of Computer Science
University of Exeter
Exeter, EX4 4RN
pw384@exeter.ac.uk

**Ke Li**
Department of Computer Science
University of Exeter
Exeter, EX4 4RN
k.li@exeter.ac.uk

## Abstract

Deep neural networks (DNNs) have demonstrated vulnerabilities to adversarial examples, which raises concerns about their reliability in safety-critical applications. While the majority of existing methods generate adversarial examples by making small modifications to the entire image, recent research has proposed a practical alternative known as adversarial patches. Adversarial patches have shown to be highly effective in causing DNNs to misclassify by distorting a localized area (patch) of the image. However, existing methods often produce clearly visible distortions since they do not consider the visibility of the patch. To address this, we propose a novel method for constructing adversarial patches that approximates the appearance of the area it covers. We achieve this by using a set of semi-transparent, RGB-valued circles, drawing inspiration from the computational art community. We utilize an evolutionary strategy to optimize the properties of each shape, and employ a simulated annealing approach to optimize the patch's location. Our approach achieves better or comparable performance to state-of-the-art methods on ImageNet DNN classifiers while achieving a lower $l_2$ distance from the original image. By minimizing the visibility of the patch, this work further highlights the vulnerabilities of DNNs to adversarial patches.

## 1 Introduction

Deep neural networks (DNNs) have revolutionized the field of computer vision, demonstrating significant progress in several tasks [38, 51, 53]. Nevertheless, they are not without vulnerabilities. Recent studies highlight a critical weakness: susceptibility to adversarial examples, where subtle, intentionally designed perturbations to input images result in the DNNs misclassification [22, 54, 43]. The existence of these adversarial examples in the physical world poses a significant threat to security-critical applications such as autonomous vehicles and medical imaging [31, 56, 33, 5]. As a result, developing methods to generate adversarial images has emerged as a critical research area for assessing the robustness of DNNs [3].

While initial studies emphasized the creation of adversarial examples with $l_p$-norm ($p$ can be 1, 2, or $\infty$) constrained perturbations, current research has shifted towards generating sparse perturbations, which alter only a small portion of the original image [12, 19, 15, 61]. These sparse perturbations have proven to be as effective as traditional $l_2$ or $l_\infty$-constrained perturbations. Adversarial patches, localized perturbations affecting a small portion of the image, have garnered significant interest as a type of sparse perturbation [47, 66, 33]. Various methods for generating adversarial patches have been proposed for both white-box (where complete information about the model is known) and black-box (where only the input-output pairs are accessible) scenarios [29, 7, 47, 15, 66, 20, 28, 11, 27].

37th Conference on Neural Information Processing Systems (NeurIPS 2023).

Proposed method                                    Patch-RS

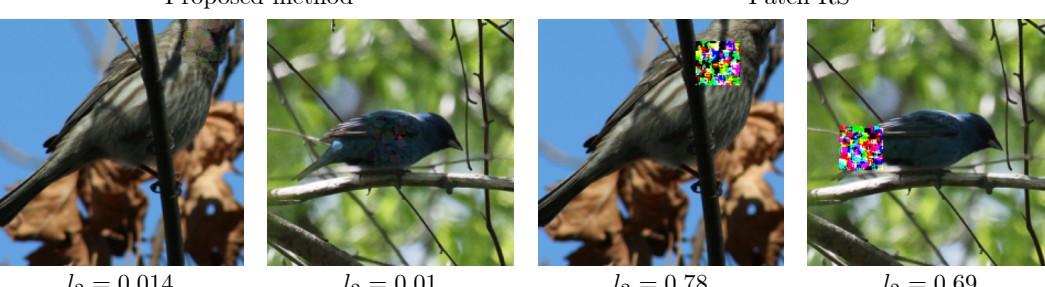

$l_2 = 0.014$       $l_2 = 0.01$       $l_2 = 0.78$       $l_2 = 0.69$

Figure 1: This illustration shows adversarial images generated by two algorithms, the proposed method and Patch-RS [15], both attacking a conventionally trained (left) and adversarially trained (right) ImageNet classifiers. While both images are adversarial, the adversarial patch generated by the state-of-the-art Patch-RS algorithm visibly distorts the image, whereas the proposed method's adversarial patch remains more similar to the original image. This similarity is demonstrated by calculating the $l_2$ distance between the adversarial patches and the area of the original image they are placed upon.

However, a significant challenge remains: the unbounded magnitudes of adversarial patches often lead to noticeable distortions in the original image, as depicted in Figure 1.

A contrasting approach to adversarial attacks is embodied by minimum-norm attacks, a class of strategies that generate adversarial examples through minimizing a particular norm of the adversarial perturbation [40, 6, 13, 58]. Due to their ability to measure adversarial accuracy under a range of perturbation budgets, these attacks serve as valuable tools for assessing DNN robustness [45]. However, these methods come with a notable drawback: they rely heavily on numerous DNN queries to substantially decrease the perturbation size. This dependence becomes particularly problematic in black-box scenarios, where the number of queries is often restricted, thus making it more difficult to reduce the perturbation size effectively [26].

Although adversarial patches have proven effective in causing DNN misclassification, existing methods often overlook the necessity of minimizing the visual impact of the patch on the original image. This oversight leads to patches that are easily detectable. To address this shortcoming, we introduce a novel method for generating adversarial patches. Our approach utilizes semi-transparent circles with RGB values, which blend into the original image. We employ simulated annealing for optimal location selection and implement an evolutionary strategy to fine-tune each circle's properties. The goal of our method is to induce the desired misclassification while minimizing the $l_2$ distance between the patch and the original image, thereby camouflaging the patch effectively.

The rest of this paper is organized as follows. Section 2 overviews some related works, underscoring their contributions and limitations. In Section 3, we outline our proposed attack scenario and provide an in-depth explanation of our method's implementation. Empirical results are presented and analyzed in Section 4. Section 5 concludes this paper and sheds some light on future directions.

## 2   Related Works

Adversarial attacks on DNNs have been one of most active fields in the machine learning community. A common strategy in these attacks is to create imperceptible perturbations, often achieved by constraining the perturbations using the $l_p$-norm. These perturbations are typically generated differently depending on the access level to the targeted DNN's information. In a white-box scenario, where the attacker has full access to the DNN's details, approaches often leverage the gradient information of the DNN's loss function. This gradient is used within an optimization algorithm to generate adversarial perturbations [54, 22, 39]. On the other hand, in a black-box scenario, where the attacker's access is limited to the DNN's output probabilities, various attack methods estimate the loss function's gradient and use it within a gradient-based optimization technique [26, 4, 59]. Some researchers have also proposed heuristic methods for the black-box scenario that do not rely on gradient information [1, 2, 27, 28, 11].

Our work fits into this landscape by focusing on the black-box scenario. However, unlike many existing methods, we aim to create adversarial patches that are not only effective but also visually blend into the original image.

## 2.1 Adversarial Patches

Adversarial patches, designed to cause image misclassification, represent a unique approach to adversarial attacks. These patches are typically small, visible, and often square-shaped, strategically applied to the targeted image [15, 47, 66]. The pioneering work by Brown *et al.* introduced the concept of universal adversarial patches, which cause misclassification when applied to a variety of images. Using the gradient information of the DNN, they utilized stochastic gradient descent to optimize the patch pattern, which was subsequently superimposed onto a set of target images at predetermined locations. Following this, Karmon *et al.* proposed LaVAN that also generates universal patches. However, they used random search to optimize the patch location. In black-box scenarios, Brown *et al.* produced universal adversarial patches by executing white-box attacks on four DNN classifiers and transferring the results to an unseen classifier. Croce *et al.* proposed the Patch-RS method, which generates adversarial patches by minimizing the aggregated loss of a set of images, using random search for patch location optimization and the Square-Attack method of Andriushchenko *et al.* for patch pattern optimization.

In contrast to universal adversarial patch approaches, other researchers have focused on generating image-specific patches. Fawzi and Frossard created patches for individual images by optimizing the position and shape of rectangular patches with a predefined monochrome pattern. Built upon the LaVAN concept, Rao *et al.* proposed alternative techniques for patch location optimization using random search variants. Yang *et al.* and Croce *et al.* further advanced the image-specific scenario, with the former using reinforcement learning for patch generation and the latter applying the Patch-RS method to minimize the loss of a single image.

Despite recent advancements in creating both universal and image-specific adversarial patches, the glaring distortions from significant modifications to input images raise practical concerns. This issue also impacts the accurate assessment of DNN robustness against adversarial patches.

## 2.2 Minimum-Norm Attacks

Minimum-norm attacks diverge from the traditional adversarial attacks by focusing on finding the smallest perturbation that can lead to misclassification under a specific norm. These attacks offer a more comprehensive assessment of DNN robustness [45]. Although white-box attacks have made significant progress in enhancing the query efficiency of minimum-norm attacks [45, 41, 48], black-box attacks still demand a substantial query budget to achieve effectiveness. The ZOO algorithm [8] constructs the problem as an aggregated function of the perturbations $l_2$-norm and weighted loss function. Estimating its gradient using finite-differences, the authors make use of coordinate descent to minimize the formulated problem. Tu *et al.* addressed the query inefficiency of ZOO by reducing the size of the perturbation using a trained Auto-Encoder DNN. Ilyas *et al.* remove the need for estimating coordinate-specific gradients by making use of natural evolutionary strategies, reducing the $l_\infty$-norm of the perturbation by iteratively maximizing the adversarial criterion within a given norm constraint, then reducing the norm. Despite the efficiency improvement of gradient estimation, existing black-box methods still require large query budgets. The SimBA method of [23] incrementally adds scaled orthogonal vectors to the current perturbation, increasing its $l_2$-norm, but is unable to reduce the $l_2$-norm of the perturbation once the desired misclassification has been achieved.

Therefore, while recent works have achieved large performance gains within the white-box scenario, black-box methods suffer from query inefficiency which restricts their applicability to real-world scenarios, particularly when the query budget is limited.

## 2.3 Evolutionary Strategies for Adversarial Attacks

Many existing studies employ evolutionary strategies (ES) for non-patch-based adversarial attacks. In the black-box scenario, ES has gained popularity due to its independence from gradient information. Notable examples include the works of [1, 46, 36, 63], who utilize evolutionary algorithms to create $l_\infty$ constrained adversarial perturbations. For conducting sparse adversarial images Williams and Li

**Algorithm 1:** Evolutionary Strategy for Generating Disguised Adversarial Patches (CamoPatch)

---

**Input:** Margin loss $\mathcal{L}$, input $\mathbf{x} \in \mathcal{X} \subseteq [0,1]^{h \times w \times 3}$, query budget $K$, sparsity $\epsilon$, initial
temperature $t$, number of circles $N$, location schedule $li$, evolutionary step-size $\sigma$

---

1   $s \leftarrow \sqrt{\epsilon}$    // Patch Side Length
2   $\boldsymbol{\delta} \leftarrow InitialPatch(N, s)$
3   $i \sim \mathcal{U}(\{0, \cdots, w - s\})$
4   $j \sim \mathcal{U}(\{0, \cdots, h - s\})$
5   $\mathbf{x}^* \leftarrow \mathbf{x}$
6   $\mathbf{x}^*_{i:i+s,j:j+s} \leftarrow \boldsymbol{\delta}$    // Apply patch
7   $L \leftarrow \mathcal{L}(\mathbf{x}^*)$
8   $norm \leftarrow ||\mathbf{x}_{i:i+s,j:j+s} - \boldsymbol{\delta}||_2$
9   **for** $k \leftarrow 1; k < K; k \leftarrow k + 1$ **do**
10     **if** $mod(k, li + 1) = 0$ **then**
11       $i, j, L, norm \leftarrow LocationUpdate()$ // see Algorithm 4
12     **else**
13       $\boldsymbol{\delta}, L, norm \leftarrow PatchUpdate()$ //see Algorithm 3

14   **return** $\mathbf{x}^*$

---

make use of a multi-objective evolutionary algorithm to minimize both the number of modified pixels and magnitude of the pixels modification. ESs have also been used to construct adversarial examples within other domains such as natural language processing [68, 67].

The use of evolutionary algorithms has also been explored for constructing adversarial patches. Chen *et al.* addressed the more-limited decision-only setting (where only the predicted label is accessible to the attacker) by placing localised regions of an image from the target class onto the attacked image. Under the constraint of the patch causing misclassification, the authors optimised the coordinates of the patch by using an adapted differential evolution algorithm to minimise the patch's $l_0$ norm.

## 3   Proposed Method

In essence, our method strives to generate adversarial patches that seamlessly blend into the targeted image by modeling the superimposed area with semi-transparent, RGB-valued circles. We adopt an approach akin to existing works that generate adversarial patches by iteratively optimizing both the patch and its position on the image. The balance between these steps is managed by a location schedule parameter, $li$. In this section, we start by defining the problem formulation, followed by a detailed description of our proposed method. The overarching structure of our method is summarized in Algorithm 1.

### 3.1   Problem Formulation

Consider a trained DNN image classifier $f : \mathcal{X} \subseteq [0,1]^{h \times w \times 3} \rightarrow \mathbb{R}^P$ which takes a benign RGB image $\mathbf{x} \in \mathcal{X}$ of height $h$ and width $w$ and outputs a label $y = \underset{p \in \{1, \cdots, P\}}{\operatorname{argmax}} f_p(\mathbf{x})$, with $P$ representing the total number of class labels. A non-targeted attack seeks a perturbation $\boldsymbol{\delta}$ satisfying:

$$\underset{p \in \{1, \cdots, P\}}{\operatorname{argmax}} f_p(\mathbf{x} + \boldsymbol{\delta}) = y_q, \tag{1}$$

where $y$ is the original class label for $\mathbf{x}$ and $y_q = \underset{q \neq y}{\operatorname{argmax}} f_p(\mathbf{x})$ is a label corresponding to a class other than the true class $y$. For targeted attacks $y_q$ is assigned a target label $y_t$, where $y_t \neq y$. In the adversarial patch scenario, the number of modified pixels is limited to maintain the semantic content of the image. Hence, the problem is cast as:

$$
\begin{aligned}
\underset{\boldsymbol{\delta}}{\text{minimize}} \quad & \mathcal{L}(f; \mathbf{x} + \boldsymbol{\delta}, y_q) \\
\text{subject to} \quad & ||\boldsymbol{\delta}||_0 \leq \epsilon, \;\; 0 \leq \mathbf{x} + \boldsymbol{\delta} \leq 1,
\end{aligned}
\tag{2}
$$

where minimizing the loss function $\mathcal{L}$ yields the desired adversarial image.

Most existing algorithms solve (2) by fixing the number of disturbed pixels to a constant value $\epsilon$, allowing unbounded modifications [15, 29, 7]. Unlike these, our proposed method aims to create adversarial patches that closely resemble the area of the original image they overlay. We approach this as a constrained optimization problem, akin to the minimum-norm setting [45, 48, 58]. Thus, our objective is to generate a $\boldsymbol{\delta}$ that solves the subsequent optimization problem:

$$
\begin{aligned}
\underset{\boldsymbol{\delta}}{\text{minimize}} \quad & ||\mathbf{x} - \boldsymbol{\delta}||_2 \\
\text{subject to} \quad & ||\boldsymbol{\delta}||_0 \leq \epsilon, \\
& \mathcal{L}(f; \mathbf{x} + \boldsymbol{\delta}, y_q) < 0, \\
& 0 \leq \mathbf{x} + \boldsymbol{\delta} \leq 1,
\end{aligned}
\tag{3}
$$

where the patch is a square with a side length of $\sqrt{\epsilon}$. We ensure that the value of the loss function $\mathcal{L}(\cdot)$ is negative when $\mathbf{x} + \boldsymbol{\delta}$ results in misclassification. This is achieved by defining the loss in the constraint as the margin loss:

$$
\mathcal{L}(f; \mathbf{x} + \boldsymbol{\delta}, y_q) = f_y - f_{y_q},
\tag{4}
$$

for non-targeted attacks and the cross-entropy loss:

$$
\mathcal{L}(f; , \mathbf{x} + \vec{\delta}, y_t) = -f_{y_t} + \log(\sum_{p=1}^{P} e^{f_p})
\tag{5}
$$

for targeted attacks.

## 3.2 Patch Initialization

We construct an adversarial patch by overlaying $N$ circles on a black image (see Figure 2), inspired by evolutionary strategies prevalent in computational art [32, 21, 57]. They aim to approximate images using semi-transparent, RGB-valued shapes. Circular shapes, due to their fewer adjustable properties, are a popular choice. Furthermore, the use of semi-transparent circular shapes have also been used by Li *et al.* and Zolfi *et al.* to construct adversarial examples. Whereas this work constructs adversarial patches, Li *et al.* and Zolfi *et al.* simulate stickers placed over a camera, modifying the entire image.

The adversarial patch $\boldsymbol{\delta}$ is represented as the concatenation of $N$ shapes:

$$
\boldsymbol{\delta} = \boldsymbol{\delta}^1 \oplus \boldsymbol{\delta}^2 \cdots \oplus \boldsymbol{\delta}^N,
\tag{6}
$$

where $\oplus$ denotes the concatenation operator. Each shape $\boldsymbol{\delta}^a$, where $a \in \{1, \cdots, N\}$, is represented by a vector comprised of seven elements including the center's coordinates $(c_1^a, c_2^a)$, the radius $r^a$, the RGB values $(R^a, G^a, B^a)$, and the shape's transparency $T^a$. These elements, normalized to continuous values between $0$ and $1$, are initially randomly sampled from a uniform distribution $\boldsymbol{\delta}^a \sim \mathcal{U}(0, 1)$. The initial location of a patch is also randomly and uniformly sampled from the available pixel locations.

## 3.3 Patch Optimization

In this paper, we employ a single solution evolutionary strategy, known as $(1 + 1)$-ES, to modify the properties of each shape $\boldsymbol{\delta}^a$. This approach has proven to be efficient for approximating images [42]. To adjust the properties, we sample values from a normal distribution $\sigma \cdot \mathcal{N}(0, I)$, where $\sigma$ is a tunable parameter controlling the trade-off between exploration (i.e., searching new areas in the solution space) and exploitation (i.e., refining the current solution). A larger $\sigma$ promotes exploration, while a smaller one favors exploitation. We then use the updated perturbation $\boldsymbol{\delta}^*$ to construct an adversarial image $\mathbf{x}^{**}$. The solution that satisfies the constraint as per (3) is retained. If both solutions meet this constraint, we opt for the one with a smaller $l_2$ distance from the original image. The patch update method is detailed in Algorithm 3 in the supplementary document.

## 3.4 Location Optimization

Addressing the discrete nature of pixel locations, many existing methods have employed random search to optimize the position of the patch within the image [15, 29, 47]. However, random search methods often falter when encountering local optima. To mitigate this, Skiscim and Golden introduced simulated annealing, a method that probabilistically accepts worse solutions based on the search *temperature* and the performance difference between current and new solutions. This approach promotes exploration of the search space in the early stages of optimization and gradually becomes more selective, favoring solutions with better quality in the later stages.

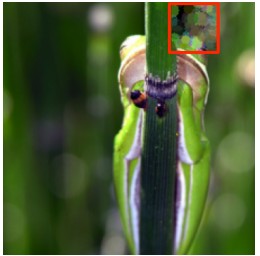 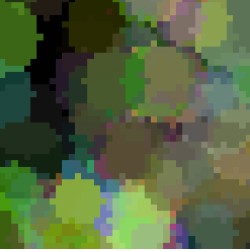

Figure 2: This illustration shows an adversarial image (left) with the adversarial patch outlined, and the magnified patch (right) for better visibility. This patch is generated by the proposed method using $N = 100$ overlapping circular shapes.

In our work, we leverage the fast simulated annealing approach proposed by Szu and Hartley to optimize the location of a patch. During each iteration $k$, we uniformly sample a single location (denoted as $(i^*, j^*)$) from the location space. Then, we apply the patch to the new location on the input image $\mathbf{x}$, and construct an updated adversarial image $\mathbf{x}^{**}$. The new solution $\mathbf{x}^{**}$ is then evaluated using the loss function $\mathcal{L}$. If both $\mathbf{x}^*$ and $\mathbf{x}^{**}$ satisfy the loss $\mathcal{L}$ constraint as per (3), we retain the solution with the lowest $l_2$-norm from the original image. Otherwise, simulated annealing is employed to probabilistically decide the acceptance of the new solution. Specifically, the acceptance probability is defined as $\exp(-d/t_{curr})$, where $d$ is the loss difference between the current and new solution, and $t_{curr} = t/k$ follows an exponentially decreasing schedule. This formulation ensures better solutions are always selected, while solutions with relatively poor quality are more likely to be accepted in the early search stages for enhanced exploration. The parameter $t$ is predefined, with larger values promoting exploration during a longer portion of the attack process. The detailed location update method can be found in Algorithm 4 in the supplementary document.

## 4 Empirical Study

In this section, we empirically evaluate our proposed method's effectiveness by attacking classifiers trained on the ImageNet dataset [16]. The experimental setup is outlined in Section 4.1, followed by a comparative analysis with state-of-the-art adversarial patch methods, including Patch-RS [15], TPA [66], OPA [20], Adv-Watermark [27] and a black-box adaptation of LOAP [47] in Section 4.2. Last but not the least, Section 4.3 offers an ablation study that scrutinizes the significance of various components and parameters within our proposed method.

### 4.1 Experimental Setup

**Dataset and Classifiers Settings:** For our experiments, we follow a similar setup to preceding works, conducting non-targeted and targeted attacks on DNN classifiers trained on the ImageNet dataset. We specifically target three adversarially trained and defended classifiers, namely AT-ResNet-50 [49], AT-WideResNet-50-2 [49] and PatchGuard [65], along with three conventionally trained classifiers, VGG-16 [51], ResNet-50 [24] and ViT-B/16 [17]. A subset of $1,000$ images, correctly classified by each classifier from the ImageNet validation set, is chosen and resized to dimensions of $224 \times 224 \times 3$. For targeted attacks we randomly select $y_t$ for each image ensuring it is different from the images true label $y$. The adversarially trained and defended classifiers are implemented using the RobustBench library [14] and authors original implementations, respectively, while the three conventional classifiers are derived from their pre-trained versions available in the PyTorch library [44]. All experiments were carried out on a system with an NVIDIA GeForce RTX 2080Ti GPU.

**Parameter Settings:** To select the value of $\epsilon$, we follow the approach of Croce *et al.*, setting $\epsilon = 1600$. This corresponds to a patch size of $40 \times 40$, which constitutes roughly $3.2\%$ of the total pixel count. We assign a budget of $10,000$ queries for each attack. As discussed in Section 3, our proposed method entails four free parameters: $\sigma$, $lit$, $t$, and $N$. For these parameters, we set $\sigma = 0.1$,

Table 1: Table presents the before and after-accuracy of each method along with the $l_2$ distance of the adversarial patch and the non-normalised residual (NNR) between the adversarial and original image after conducting non-targeted attacks. We provide the mean and variance of each metric over 10 runs.

| Attack Method | AT-WideResNet-50-2 | | | AT-ResNet-50 | | |
|---|---|---|---|---|---|---|
| | Accuracy | $l_2$ | NNR | Accuracy | $l_2$ | NNR |
| - | 68.46% | - | - | 64.02% | - | - |
| CamoPatch | **12.98% (0.01)**† | **0.14 (0.05)**† | **0.12 (0.07)**† | **6.00% (0.03)**† | **0.15 (0.03)**† | **0.13 (0.03)**† |
| Patch-RS* | 14.42% (0.01)‡ | 0.43 (0.07)‡ | 0.30 (0.05)‡ | 12.00% (0.02)‡ | 0.41 (0.12)‡ | 0.33 (0.05)‡ |
| Patch-RS | 14.42% (0.01)‡ | 0.74 (0.08)‡ | 0.42 (0.07)‡ | 12.00% (0.02)‡ | 0.74 (0.09)‡ | 0.43 (0.07)‡ |
| TPA | 51.66% (1.3)‡ | 0.82 (1.21)‡ | 0.82 (0.07)‡ | 34.82% (1.41)‡ | 0.92 (0.05)‡ | 0.87(0.09)‡ |
| OPA | 36.88% (0.1)‡ | 0.76 (0.20)‡ | 0.74 (0.05)‡ | 24.83% (1.12)‡ | 0.77 (0.14)‡ | 0.75 (0.04)‡ |
| LOAP | 38.85% (0.4)‡ | 0.56 (0.02)‡ | 0.46 (0.03)‡ | 48.89% (0.1)‡ | 0.72 (0.18)‡ | 0.64 (0.03)‡ |
| Adv-watermark | 52.00% (0.3)‡ | 0.37(0.05)‡ | 0.23(0.07)‡ | 44.00% (0.3)‡ | 0.42 (0.02)‡ | 0.29 (0.07)‡ |

| Attack Method | ViT-B/16 | | | BagNet9 with PatchGuard | | |
|---|---|---|---|---|---|---|
| | Accuracy | $l_2$ | NNR | Accuracy | $l_2$ | NNRl |
| - | 77.91% | - | - | 55.1% | - | - |
| CamoPatch | **8.00% (0.05)**† | **0.09 (0.02)** | **0.12 (0.02)** | **3.20% (0.01)**† | **0.07(0.03)**‡ | **0.11 (0.01)**‡ |
| Patch-RS* | 19.00% (0.10)‡ | 0.68 (0.05)† | 0.39 (0.07)‡ | 5.80% (0.02)‡ | 0.42 (0.05)‡ | 0.30 (0.05)† |
| Patch-RS | 19.00% (0.10)‡ | 0.71 (0.12)† | 0.41 (0.09)‡ | 5.80% (0.02)‡ | 0.62 (0.18)‡ | 0.57 (0.11)† |
| TPA | 38.12% (0.91)‡ | 0.59 (0.08)‡ | 0.54 (0.09)‡ | 32.87% (1.45)‡ | 0.62 (0.11)‡ | 0.61(0.09)‡ |
| OPA | 33.09% (0.17)‡ | 0.68 (0.23)‡ | 0.68 (0.07)‡ | 57.89% (2.01)‡ | 0.61 (0.16)‡ | 0.67 (0.04)‡ |
| LOAP | 43.91% (0.80)‡ | 0.63 (0.05)‡ | 0.50 (0.13)‡ | 72.82% (0.14)‡ | 0.89 (0.23)‡ | 0.78 (0.11)‡ |
| Adv-watermark | 36.01% (0.12)‡ | 0.17(0.04)‡ | 0.28(0.03)‡ | 42.00% (0.45)‡ | 0.14(0.01)‡ | 0.29(0.05)‡ |

| Attack Method | VGG-16 | | | ResNet-50 | | |
|---|---|---|---|---|---|---|
| | Accuracy | $l_2$ | NNR | Accuracy | $l_2$ | NNR |
| - | 73.36% | - | - | 76.12% | - | - |
| CamoPatch | 9.70% (0.03) | **0.09 (0.02)**† | **0.11 (0.02)**† | **10.00% (0.02)**† | **0.08 (0.01)**† | **0.10 (0.01)**† |
| Patch-RS* | **6.82% (0.04)** | 0.42 (0.02)‡ | 0.30 (0.05)‡ | 15.92% (0.02)‡ | 0.45 (0.04)‡ | 0.31 (0.04)‡ |
| Patch-RS | **6.82% (0.04)** | 0.63 (0.01)‡ | 0.61 (0.07)‡ | 15.92% (0.02)‡ | 0.67 (0.08)‡ | 0.69 (0.07)‡ |
| TPA | 47.11% (1.30)‡ | 0.61 (0.13)‡ | 0.55 (0.05)‡ | 38.98% (1.41)‡ | 0.61 (0.07)‡ | 0.58(0.07)‡ |
| OPA | 32.19% (0.10)‡ | 0.71 (0.20)‡ | 0.64 (0.06)‡ | 27.91% (1.12)‡ | 0.71 (0.14)‡ | 0.66 (0.04)‡ |
| LOAP | 37.99% (0.40)‡ | 0.68 (0.02)‡ | 0.63 (0.05)‡ | 47.99% (0.10)‡ | 0.78 (0.12)‡ | 0.67 (0.05)‡ |
| Adv-watermark | 32.00% (0.10)‡ | 0.13(0.08)‡ | 0.25(0.05)‡ | 35.00% (0.40)‡ | 0.16(0.01)‡ | 0.31(0.07)‡ |

† denotes the performance of the method significantly outperforms the compared methods according to the Wilcoxon signed-rank test [60] at the 5% significance level; ‡ denotes the corresponding method is significantly outperformed by the best performing method (shaded).

$t = 300$, $lit = 4$, and $N = 100$. We provide an empirical justification for these specific settings in Section 4.3.

**Performance Metrics:** We evaluate the performance of all considered algorithms by allowing each method to exhaust the allocated query budget while attacking each classifier. To evaluate the effectiveness of an attack we report the accuracy of the classifier on the generated adversarial images. For the successful adversarial images, we report two additional metrics: (1) the $l_2$ distance between the adversarial patch and the corresponding area of the original image, and (2) the non-normalised residual (NNR) between the adversarial and original image, which measures the absolute difference between the pixel values of the constructed patch and the area of the original image it covers.

Given the stochastic nature of our proposed method and the comparison methods, we follow the setup of [15] and report the mean and variance of each metric over 10 independent runs with different random seeds. We additionally utilize the Wilcoxon signed-rank test [60] at a 5% significance level to statistically verify whether the improvements by our method over the compared algorithms across the 10 runs are significant.

## 4.2 Comparison

For the adaptation of LOAP [47], we replace its gradient computation method with the estimation method of [26]. The detailed description of this estimation method can be found in Algorithm 2 in the supplementary document. Additionally, we compare our method with an adapted version of Patch-RS, where we minimize the $l_2$ distance of the constructed patch in a similar manner to our proposed method. For each compared algorithm, we utilize the authors' original implementation and recommended settings. In our black-box adaptation of LOAP [47], we set the number of iterations $n = 50$ and variance $\eta = 0.001$ for the gradient estimation method of Ilyas *et al.*.

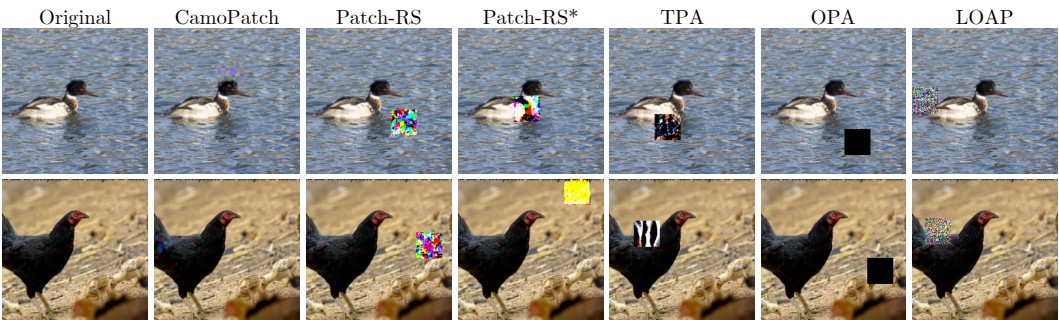

Figure 3: Adversarial images generated by methods conducting non-targeted attacks on the conventionally trained VGG-16 (top) and adversarial trained WideResNet-50-2 [49] (bottom) classifiers. Whereas adversarial patches generated by state-of-the-art methods are visibly clear, the patches generated by the proposed method are well camouflaged within the image.

**Results:**    Table 1 present the statistical results of non-targeted attacks conducted on the trained ImageNet classifiers. In the tables, "CamoPatch" denotes our proposed method, and "Patch-RS*" refers to the adapted Patch-RS algorithm.

These results demonstrate that the Patch-RS attack, along with our own method, achieves higher attack success rates compared to the other state-of-the-art methods. This result aligns with previous work [15], which demonstrated the superior performance of the Patch-RS algorithm. Despite Patch-RS outperforming our method when attacking the VGG-16 classifier, according to the Wilcoxon signed-rank test, there is no significant difference between the performance of both methods. Alternatively, when attacking the remaining five classifiers, the proposed method is able to significantly outperform Patch-RS and other compared methods according to the Wilcoxon signed-rank test.

Comparing the $l_2$ distance and NNR of adversarial patches generated by the attack methods, the proposed method is able to construct adversarial patches that are far less invasive to the input image. This is supported by the proposed method significantly outperforming all other methods in terms of both $l_2$ distance and NNR, according to the Wilcoxon signed-rank test. This result highlights that the effectiveness of our adversarial patches is not compromised by their perceptibility.

Despite the adapted Patch-RS* algorithm being able to generate patches with lower $l_2$ distances from the original image compared to its original implementation, its use of Square-Attack [2] for patch pattern optimization results in the patch values taking the corners of the color cube $[0, 1]$. Therefore, its ability for $l_2$ minimization is significantly hampered. Alternatively, the proposed method is able to construct patches with any color, which allows for effective approximations of the original image area. Figure 3 provides a visual comparison of adversarial images generated by each method when attacking the VGG-16 and AT-ResNet-50 classifiers.

We report the results of the targeted attacks in Section 6.1 of the supplementary material.

**Robustness Evaluation:**    Despite the assumption that adversarial trained classifiers have improved robustness compared to their conventionally trained counterparts, our experimental results reveal a different picture. The proposed method achieves higher success rates when attacking the adversarial trained classifier AT-ResNet-50 of Salman *et al.* compared to the conventionally trained VGG-16 and ResNet-50 classifiers.

However, the results also demonstrate that the adversarial patches generated by the proposed method, when attacking the AT-ResNet-50 classifier, exhibit larger $l_2$ distances from the original image, resulting in larger non-normalised residuals between the entire adversarial image and the original image. This suggests that while the adversarial trained classifier is more susceptible to adversarial patches, these patches require larger distortions to cause the misclassification. On the other hand, conventionally trained classifiers are more susceptible to smaller changes in the original image. This behavior can be attributed to the general procedure of adversarial training, which introduce noise onto images during the training to enhance robustness. Consequently, patches with larger impact decrease the likelihood of the image being representative of the training data, as has been observed in other works [50]. Furthermore, wee see the AT-WideResNet-50-2 classifier exhibits greater robustness to

Table 2: Table presents the before and after-accuracy of each CamoPatch configuration along with the $l_2$ distance of the adversarial patch and the non-normalised residual (NNR) between the adversarial and original image after conducting non-targeted attacks. We provide the mean and variance of each metric over 10 runs.

| | | | | VGG-16 | | | |
|---|---|---|---|---|---|---|---|
| $li$ | $t$ | $N$ | $\sigma$ | Accuracy | $l_2$ | NNR | Runtime(s) |
| - | - | - | - | 76.12% | - | - | - |
| 1 | 300 | 100 | 0.1 | 12.88%(1.0) | **0.09(0.04)** | 0.14(0.01) | 440.03(10.32) |
| 4 | 100 | 100 | 0.1 | 10.79%(1.5) | **0.09(0.07)** | 0.13(0.02) | 440.01(10.05) |
| 4 | 300 | 100 | 0.1 | **9.64%(1.0)** | **0.09(0.05)** | **0.11(0.02)** | 440.03(10.32) |
| 4 | 300 | 100 | 0.3 | 11.66%(2.0) | **0.09(0.06)** | **0.11(0.05)** | **439.13(10.56)** |

[†] denotes the performance of the method significantly outperforms the compared methods according to the Wilcoxon signed-rank test [60] at the 5% significance level; [‡] denotes the corresponding method is significantly outperformed by the best performing method (shaded).

our method. Since both AT-WideResNet-50-2 and AT-ResNet-50 models are trained using the same process, these results suggest that the WideResNet architecture is inherently more robust.

### 4.3 Ablation Study

The proposed method consists of four tunable parameters: $li, N, \sigma$, and $t$. To determine their optimal values, we conduct a grid search over the parameter space. Specifically, we explore $li \in \{1, 4\}$, $N \in \{100, 300\}$, $t \in \{100, 300\}$, and $\sigma \in \{0.1, 0.3\}$. The choice of $li$ follows the recommendation of Croce *et al.*, while the values of $\sigma$, $t$, and $N$ are commonly used in the evolutionary[52] and computational art [57] communities, respectively.

To evaluate the performance of each parameter configuration, we conduct non-targeted attacks on the VGG-16 ImageNet classifier using 1000 correctly classified images from the validation set. We measure the accuracy of the model on the generated adversarial images, $l_2$ distance and NNR for each configuration over 10 independent runs with different random seeds. Additionally, we compare the computational time required for each configuration to complete an attack on a single image.

**Configurations:** Table 2 presents the four top performing configurations in terms of the attack accuracy. The results demonstrate that the performance of the proposed method heavily depends on the number of circles $N$ used to construct the patch pattern. Increasing the number of circles allows for better detailed approximations but also introduces additional complexity. From the results in Table 2, we observe that the best performing configurations all use $N = 100$, suggesting that the patch optimizer, $(1 + 1)$-ES, struggles with larger numbers of circles. Moreover, we observe longer runtimes for $N = 300$ due to the increased number of properties that need adjustment. Beyond the number of circles $N$, the proposed method achieves improved performance with a larger budget for patch pattern optimization ($li = 4$) and a larger exploration parameter for location optimization ($t = 300$). Based on these findings, we set the optimal parameter configuration of the proposed method to $li = 4, t = 300, N = 100$, and $\sigma = 0.1$.

**Simulated Annealing:** To justify the use of simulated annealing for location optimization within the proposed method, we compare its performance with and without simulated annealing. Removing simulated annealing results in a pure random search method similar to existing works. We keep the other parameters of the proposed method constant with those outlined in Section 4.1. The results in Table 3 demonstrate the improved performance exhibited by the proposed method when the simulated annealing policy is employed for location optimization, particularly when attacking adversarial trained classifiers. Despite generating patches with a higher $l_2$ distance, the proposed method with simulated annealing achieves higher success rates. This suggests that more challenging images require larger distortions to cause misclassification, increasing the average $l_2$ distance of the generated successful adversarial patches.

## 5 Contributions, Limitations and Future Work

**Contributions:** In this work, we propose CamoPatch, a novel attack method for generating adversarial patches that can blend into the targeted image. We achieve this by constructing the patch

Table 3: Table presents the before and after-accuracy of the CamoPatch method with (CamoPatch) and without (CamoPatch*) the simulated annealing policy for location optimization, along with the $l_2$ distance of the adversarial patch and the non-normalised residual (NNR) between the adversarial and original image after conducting non-targeted attacks. We provide the mean and variance of each metric over 10 runs.

| Classifier | CamoPatch | | | CamoPatch* | | |
|---|---|---|---|---|---|---|
| | ASR | $l_2$ | NNR | ASR | $l_2$ | NNR |
| VGG-16 | **90.30% (0.03)** | **0.09 (0.02)** | **0.11 (0.02)** | 90.01 (0.1) | **0.09 (0.01)** | **0.11 (0.02)** |
| ResNet-50 | **90.00% (0.02)** | **0.08 (0.01)** | **0.08 (0.01)** | **90.00% (0.01)** | 0.09 (0.01) | 0.1 (0.02) |
| AT-WideResNet-50-2 | **87.02% (0.01)**[†] | 0.14 (0.05)[‡] | **0.12 (0.07)** | 83.02% (0.02)[‡] | **0.11 (0.05)**[†] | **0.09 (0.02)** |
| AT-ResNet-50 | **94.00% (0.03)**[†] | 0.15 (0.03) | **0.13 (0.03)** | 90.00% (0.01) | **0.14 (0.03)** | **0.13 (0.05)** |

[†] denotes the performance of the method significantly outperforms the compared methods according to the Wilcoxon signed-rank test [60] at the 5% significance level; [‡] denotes the corresponding method is significantly outperformed by the best performing method (shaded).

pattern using a combination of semi-transparent, RGB-valued circles, which are optimized to cause misclassification and approximate the covered area of the original image. By incorporating a simulated annealing policy for location optimization, our method generates adversarial patches with improved or comparable success rates, while minimizing the visual impact on the target image.

**Ethical Considerations:** Adversarial patches have gained attention due to their potential real-world applications, where attackers can print and physically place them to deceive real-world implemented DNNs [7, 18]. However, existing methods often generate patches that are visually clear and easily detectable to a human observer. Our work introduces the concept of camouflaged adversarial patches, which are difficult for both humans and computer vision systems to perceive. This raises further concerns about the robustness of DNN classifiers in safety-critical applications. Adversarial training has proven to be an effective method of improving the robustness of DNN classifiers to adversarial patches [47]. Incorporating images with camouflaged adversarial patches into the training process of DNNs may be a promising avenue to enhance their robustness and mitigate the vulnerabilities demonstrated in this work.

**Limitations and Future Work:** It is important to acknowledge the limitations of the proposed method and identify potential areas for future research. One limitation is that our method assumes the attacker has access to the output probabilities of the targeted DNN, which may not always be the case in real-world scenarios. Future work could explore adapting the proposed method to scenarios where only the predicted label of the input is available, by utilizing estimated loss functions such as the one proposed by Ilyas *et al.*. Furthermore, techniques from weakly supervised learning can be incorporated into the proposed method to address the label-only setting. Specifically, by using estimation techniques [26] to score constructed adversarial images, the use of weakly supervised image classification models [25] as surrogates could improve the efficiency of the proposed method in addition to providing a direction of the search for the label-only setting. Alternatively, utilizing the stochastic nature of the proposed method to generate a set of non-evaluated candidate solutions, the use of the weakly supervised learning techniques such as semi-supervised learning could be applied [30] to train a surrogate model on both evaluated and non-evaluated adversarial images. Thereby using the surrogate to select predicted-optimal solutions for evaluation.

Another limitation is the DNN query budget assumption in our experiments. In practice, the available query budget might be significantly lower. To address this limitation, future research could extend the proposed method to incorporate surrogates or approximation models that guide the attack process, using fewer DNN queries, similar to that of Williams *et al.* within the computational art field.

Lastly, the parameter tuning process in our work follows a conventional grid-search approach, which limits the exploration of parameter combinations. Bayesian optimization methods could be employed to automate the configuration of attack algorithms, leveraging Gaussian Process surrogate models to handle the stochastic nature of the proposed method and guide the parameter search [34, 10, 9, 37]. This would provide a more efficient approach for determining the optimal parameter configuration.

## Acknowledgement

This work was supported in part by the UKRI Future Leaders Fellowship under Grant MR/S017062/1 and MR/X011135/1; in part by NSFC under Grant 62376056 and 62076056; in part by the Royal Society under Grant IES/R2/212077; in part by the EPSRC under Grant 2404317; in part by the Kan Tong Po Fellowship (KTP/R1/231017); and in part by the Amazon Research Award and Alan Turing Fellowship.

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
