# OpenReview forum: "CamoPatch: An Evolutionary Strategy for Generating Camoflauged Adversarial Patches"
_NeurIPS.cc/2023/Conference — NeurIPS 2023 poster_

### Official Review · Reviewer_sS1s · 2023-07-02

**Soundness:** 3 good
**Presentation:** 3 good
**Contribution:** 3 good
**Rating:** 5
**Confidence:** 4

**Summary:**

Existing methods often produce clearly visible distortions since they do not consider the visibility of the patch. To address this, the authors propose a novel method for constructing adversarial patches that approximates the appearance of the area it covers. They achieve this by using a set of semi-transparent, RGB-valued circles, drawing inspiration from the computational art community. They utilize an evolutionary strategy to optimize the properties of each shape, and employ a simulated annealing approach to optimize the patch’s location.

**Strengths:**

1. The research content of the  paper is meaningful and important
2. The paper is well written
3. Experiments are relatively abundant

**Weaknesses:**

Although the method proposed in this paper offers some practical applications, its level of innovation is relatively limited, resulting in incremental improvements.

**Questions:**

1. The use of the l_2 distance evaluation index may not be suitable for non-patch attacks, as it primarily focuses on measuring the pixel-level differences between the original and perturbed images. As for the experimental comparison algorithm, it would be valuable to investigate if it includes attacks that introduce overall image disturbance, such as global modifications or image-level perturbations. This would provide a more comprehensive evaluation of the algorithm's robustness against different types of attacks.
2. The number of images chosen for the ablation experiment is relatively small, consisting of only a hundred images. This limited sample size may impact the reliability and generalizability of the experiment's results, especially if the observed effects are not clearly evident. A larger and more diverse set of images would provide a more robust evaluation of the proposed method.
3. The observation that random methods yield comparable results to the proposed method in this paper raises questions about the effectiveness and uniqueness of the proposed approach. Further analysis is needed to understand the reasons behind this phenomenon.
Please do more experiments to compare the random method with the proposed method and explain why they are similar in attacking.


**Limitations:**

The method presented in this paper can be considered as a heuristic black box attack algorithm that incorporates concealment during the attack process. However, it should be noted that this approach is not entirely novel and has been explored to some extent in previous research.  The authors can refer to the following papers:

 [1] Zhong Y, Liu X, Zhai D, et al. Shadows can be dangerous: Stealthy and effective physical-world adversarial attack by natural phenomenon[C]//Proceedings of the IEEE/CVF Conference on Computer Vision and Pattern Recognition. 2022: 15345-15354.

[2] Jia X, Wei X, Cao X, et al. Adv-watermark: A novel watermark perturbation for adversarial examples[C]//Proceedings of the 28th ACM International Conference on Multimedia. 2020: 1579-1587.

[3] Wei X, Guo Y, Yu J. Adversarial sticker: A stealthy attack method in the physical world[J]. IEEE Transactions on Pattern Analysis and Machine Intelligence, 2022, 45(3): 2711-2725.

---

> ### Author Rebuttal · Authors · 2023-08-09
>
> **W#1:** We argue the innovation of our work lies in its consideration of visibility to adversarial patches as well as its methodology for generating adversarial patches. Specifically, previous works do not consider the visibility of a patch resulting in clear distortions to the image as shown in Figure 1 of our manuscript. We argue that this not only questions the practicality of the attack but also the accuracy of their use for DNN robustness evaluation (lines 92 to 94). To construct the pattern of the patch, many works use previously proposed non-patch attack methods (references [12, 34, 7]) which involve the modification of pixel values either through gradient of heuristic methods. Alternatively, other methods make use of pre-defined textures and color (references [48, 16]). Different to existing methods, we construct the patch using a set of semi-transparent RGB circles, which to our knowledge has not been previously proposed. Our novel method of patch construction not only results in more effective attacks (Table 1 in manuscript, **Tables 1 and 2 of the global 'author rebuttal PDF'**) but also generates adversarial patches with low visibility (Figures 1 and 3). Furthermore, whereas random search has shown to be the go-to approach for patch location optimization (lines 169 to 170), we demonstrate our use of simulated annealing results is a more effective method for location optimization (Table 2 of our manuscript). Finally, as our method minimizes the $l_2$ distance to reduce visibility, our patch construction method can be applied to the non-patch minimum-norm attack scenario.
>
> **Q#1:** We address the reviewer's concern from the following two aspects.
> - The $l_2$ distance has been extensively used to promote or ensure low visibility of adversarial perturbation within non-patch adversarial attacks (lines 27 to 28; references [41, 18, 27, 4, 2]), either by its use for constraining the size of the perturbation (lines 58 to 60) or through its minimization (lines 100 to 102; references [33, 29, 25, 8, 19]).
> - In this work focuses on the generation of adversarial patches. Non-patch attacks consider different assumptions and constraints applied to the perturbations (lines 27 to 29; lines 58 to 61), therefore for the sake of fairness we only compare our method to patch adversarial attacks. This is also reflected in previous works (references [2, 12, 4, 48]) that compare methods with similar assumptions and constraints.
>
> **Q#2:** As requested by the reviewer, we have amended our ablation study to 1,000 correctly classified images from the ImageNet validation set. The results of the top $4$ performing configurations are reported in **Table 3 of the global 'author rebuttal PDF'**. From our experiments, we find the observation is similar to our previous setting.
>
> **Q#3:** We argue the uniqueness of our work lies in its consideration of visibility to adversarial patches as well as its methodology for generating adversarial patches. Specifically, previous works do not consider the visibility of a patch resulting in clear distortions to the image (Figure 1). We argue that this not only questions the practicality of the attack but also the accuracy of their use for DNN robustness evaluation (lines 92 to 94). To construct the pattern of the patch, many works use previously proposed non-patch attack methods (references [12, 34, 7]) which involve the modification of pixel values either through gradient of heuristic methods. Alternatively, other methods make use of pre-defined textures and color (references [48, 16]). Different to existing methods, we construct the patch using a set of semi-transparent RGB circles, which to our knowledge has not been previously proposed. This novel method of patch construction not only results in more effective attacks (Table 1 in our manuscript, as well as **Tables 1 and 2 of the global 'author rebuttal PDF'**) but also generates adversarial patches with low visibility (Figures 1 and 3 of our manuscript). Furthermore, whereas random search has shown to be the go-to approach for patch location optimization (lines 169 to 170), we demonstrate our use of simulated annealing results is a more effective method for location optimization (Table 2 of our manuscript).
>
> **Limitations.** We thank the reviewer for providing these three reference. However, we believe our proposed method is unique from the following three aspects.
> - We focus on generating adversarial patches, whereas the works raised by the reviewer aim to generate physical changes which change the scene of the target image (i.e. changes in shadows [1] or stickers placed on a sign [2]).
> - [1] to [3] do not consider the visibility of their changes, instead their changes reflect plausible realistic changes to the image. In contrast, we aim to reduce the visibility of the adversarial patch that is placed upon the target image.
> - [1] to [3] are constrained by the object chosen to be placed upon the target image (i.e. a shadow, a sticker or watermark). Differently, our proposed method constructs adversarial patches by optimizing the characteristics of a set of semi-transparent circles, allowing the patch to approximate any image it is placed upon.

---

> > ### Comment · Reviewer_sS1s · 2023-08-21
> >
> > Thanks to the author for his careful reply. The information provided by the author has effectively clarified most of my inquiries. After a thorough evaluation of the provided content and the feedback provided by other reviewers, I have decided to revise my initial rating.

---

### Official Review · Reviewer_1i2p · 2023-07-03

**Soundness:** 3 good
**Presentation:** 3 good
**Contribution:** 2 fair
**Rating:** 5
**Confidence:** 5

**Summary:**

This paper explores black-box score-based patch attacks on image classification. The previous patch attacks are relatively visible to the naked eye, so the authors hope to have a camouflaged adversarial patch. Therefore, the authors propose a novel method for constructing adversarial patches that approximates the appearance of the area it covers. This camouflaged patch is composed of multiple translucent RGB circles. With the help of evolutionary algorithm optimization, the proposed attack can optimize the shape and location of patches. The proposed method achieves better or comparable performance to state-of-the-art methods and is stealthy.

**Strengths:**

1. Camouflaged adversarial patches are a more significant threat to security than normal adversarial patches.

2. The proposed method is visually better imperceptible.

**Weaknesses:**

1. Experimentation is not sufficient. The authors only performed robustness evaluation on two CNNs, and did not test on more CNNs or updated ViT, which limits the generalization of the method;
The authors do not have HPA [1], MPA [1], and Adv-watermark [2], the performance of these methods is unknown in imperceptibility, and they are all score-based patch attacks;
The author did not test on patch defense, including LGS [3], DW [4], and PatchGuard [5].

2. The idea of a translucent RGB patch is not very novel in patch attacks and has been discussed in [6] and [7]. The differences between this work and its patch form are not discussed.

3. In the absence of experiments on targeted attacks, we believe that the color prior cannot achieve targeted attacks.

4. There is no curve of attack performance changing with the number of queries, which is of great significance to the analysis of algorithm convergence.

[1] PatchAttack: A Black-Box Texture-Based Attack with Reinforcement Learning, ECCV 2020

[2] Adv-watermark: A Novel Watermark Perturbation for Adversarial Examples, ACM MM 2020

[3] Local gradients smoothing: Defense against localized adversarial attacks, WACV 2019

[4] On visible adversarial perturbations & digital watermarking, CVPRW 2018

[5] PatchGuard: A Provably Robust Defense against Adversarial Patches via Small Receptive Fields and Masking, S&P 2021

[6] Adversarial camera stickers: A physical camera-based attack on deep learning systems, ICML 2019

[7] The Translucent Patch: A Physical and Universal Attack on Object Detectors, CVPR 2021

**Questions:**

See Weaknesses.

**Limitations:**

Yes.

---

> ### Author Rebuttal · Authors · 2023-08-09
>
> **W#1:** We address the reviewer's concerns from the following two aspects.
> - In fact, We evaluate the proposed method by conducting adversarial attacks on four, including two adversarial and two non-adversarial, classifiers.
> - In this work we compare the proposed method with the TPA algorithm of PatchAttack (reference [1] provided by the reviewer). We additionally make comparisons with the HPA algorithm used within PatchAttack but refer to it as OPA (reference [16] of our manuscript). We do not make comparisons with the MPA algorithm of PatchAttack due to its similarities with the HPA algorithm as well as their similar performance. Finally, our decision was also made by considering the use of MPA from reference [1] provided by the reviewer as motivation for the development of TPA texture-based attack (subsection 3.3 of [48]).
> - At the request of the reviewer, we show the results of conducted non-targeted attacks on a Transformer based DNN classifier (Dosovitskiy et al., An Image is Worth 16x16 Words: Transformers for Image Recognition at Scale, ICLR'21 ) in **Table 1 of the global 'author rebuttal PDF'**. We additionally amend our experiments to include the Adv-watermark algorithm whilst also attacking a PatchGuard (reviewer reference [5]) defended model, due to it being a more recent defence mechanism.
>
> **W#2:** Despite the use of translucent RGB shapes not being novel, we argue that our method for generating adversarial patches is. Specifically, the work 'adversarial camera stickers' (reference [6] provided by the reviewer) focuses on making physical changes by applying semi-transparent stickers over the camera, resulting in modifications to the entire image. Similarly, The 'translucent patch' (reference [7] provided by the reviewer) applies a similar technique to object detection networks by simulating translucent stickers placed upon the lens of a camera. Differently, our work only modify a small localized region of the image (patch) using translucent RGB circles to construct its pattern. Furthermore, the changes made by [6] and [7] to the target image are clearly visible whereas our primary goal is to minimize the visibility of changes made to the target image. We will revise our original manuscript to include the differences between the works [6, 7] and ours.
>
> **W#3:** As requested by the reviewer, we amend our experiments to include targeted attacks on both adversarial and non-adversarial trained classifiers. The results of these experiments can be found in **Table 2 of the global 'author rebuttal PDF'**.
>
> **W#4:** As requested by the reviewer, we include the performance curve of the proposed and compared algorithms in **Figure 1 of the global 'author rebuttal PDF'**.

---

> > ### Comment · Reviewer_1i2p · 2023-08-16
> >
> > The authors' explanations basically answer our questions, but there are still two questions that we don't know:
> >
> > 1. How to choose the classes of targeted attacks? We think this is important for the fairness of the experiment.
> > 2. Why is the introduction of non-normalized residual? We feel that there seems to be a lack of relevant description.

---

> > > ### Author Response · Authors · 2023-08-16
> > >
> > > We thank the reviewer for providing these comments.
> > >
> > > **Q1:** To conduct targeted-attacks we follow the same setup as previous works (reference 12 within the manuscript) and select a random target class for each image.
> > >
> > > **Q2:** After the suggestion made by **reviewer tLps (W#2)**, we amend our evaluation to include the “non-normalised residual” metric which measures the absolute difference between the pixel values of the constructed patch and the area of the original image it covers. Following the original experimental setup, we report the mean and variance over 10 independent runs, and apply the Wilcoxon signed-rank test (reference 47 within the manuscript). We will amend our evaluation to include this metric within the final version of the manuscript in addition to a definition of the metric.

---

> > > > ### Comment · Reviewer_1i2p · 2023-08-17
> > > >
> > > > The authors' response solves our problem, and the authors have released the code, so we increase the rating.

---

### Official Review · Reviewer_eiFr · 2023-07-06

**Soundness:** 2 fair
**Presentation:** 2 fair
**Contribution:** 1 poor
**Rating:** 6
**Confidence:** 3

**Summary:**

Adversarial examples for a DNN that are not visible to humans are generated with Evolutionary Strategies. The examples are semi-transparent RGB-valued circles. The shape and position are optimized.

**Strengths:**

- ES and SA are gradient free methods
- Patch detectability is taken into account


**Weaknesses:**

- How is visibility evaluated? Is L2 always working for the distance
- No defense to new attack provided
- Limited presentation of Evolutionary Computation methods for adversarial examples in the related works
- Limited discussion of computational cost

**Questions:**

Questions:
- Is \sigma used in Algorithm 1?
- What post-hoc adjustment did you use?
- Why only 10 runs?

---

> ### Author Rebuttal · Authors · 2023-08-09
>
> **W#1:** The norm of a perturbation, including $l_2$, has proven to be a reliable metric of evaluate the visibility of a perturbation, with previous works either constraining (lines 58-60; reference [2]) or minimizing (section 2.2; references [45, 33, 29, 35, 8]) its value to ensure or promote the imperceptibility of the perturbation. We additionally make use of the SSIM metric to measure the structural impact of the patch on the overall image.
>
> **W#2:** The purpose of this work is to further highlight the vulnerability of DNN image classifiers to adversarial patches by proposing a novel attack algorithm that generates adversarial patches with lower visibility compared to current state-of-the-art methods, i.e. camouflaged. Although this work does not propose a defense mechanism to our attack, we discuss possible directions to address the vulnerabilities of DNNs demonstrated by this work in lines 311 to 320 of our manuscript. In particular, incorporating camouflaged adversarial patches in the training dataset, a.k.a. adversarial training, is a promising way to defend against our proposed method.
>
> **W#3:** To the best of our knowledge, the use of evolutionary strategies (ES) have not been explored for the construction of adversarial patches, therefore our discussion of ES is limited. However, within the black-box scenario, the use of meta-heuristics such as ES has becomes a popular approach due to their independence from gradient information. Allowing all pixels of an image to be modified, particular works include that of Alzentot et al., (GenAttack, GECCO'19) who make use of an evolutionary algorithm to evolve a population of adversarial images. Additionally, the work of Qiu et al., (Black-box adversarial attacks using evolution strategies, GECCO'21) apply the popular CMA-ES algorithm to generate adversarial images. Similarly, the work of Li et al. (https://doi.org/10.1109/TEVC.2022.3151373) make use of the differential evolution optimizer to generate adversarial images. Alternatively, the use of ES have also been applied to the sparse adversarial attack scenario. Specifically, the work of Su et al. also make use of differential evolution to generate adversarial images by modifying a single pixel of an image. We will include this discussion within our revisions.
>
> **W#4:** As shown in Table 2 of our manuscript, we compare the runtime of different hyper-parameter configurations. This partially covers the evaluation of the computational cost of our proposed method. On the other hand, for the sake of fair comparisons, we fixed the computational budget to be 10,000 queries (line 208; lines 212 to 213) for all peer algorithms in our experiments. In particular, we believe the querying of the attacked deep neural network is the most computationally demanding area of an adversarial attack.
>
> **Q#1:** $\sigma$ is a parameter used within the patch optimization stage of the proposed method. It controls the trade-off between exploration and exploitation (lines 154 to 161). We describe this process in Algorithm 3 in the appendix (lines 166 to 167) which also describes the use of $\sigma$. In the revised version, we will include the arguments of both Location and Patch optimization processes within Algorithm 1 to improve the clarity of $\sigma$ and other parameters.
>
> **Q#2:** In our experiments, we have conducted a parameter tuning upon four parameters $li,N,\sigma$ and $t$ involved in our proposed method (lines 272 to 276). In particular, we used the VGG-16 ImageNet model as the victim (please refer to the ablation study, Section 4.3 of our manuscript).
>
> **Q#3:** We follow the settings used in other works (references [2,12]), as cited in lines 218 to 222.

---

> > ### Comment · Reviewer_eiFr · 2023-08-17
> >
> > Thank you to the authors for the additional efforts to improve the experimentation and answer my questions. My review will remain unchanged.

---

> > ### Comment · Reviewer_1i2p · 2023-08-17
> >
> > **To W#3:** Recent work [1] and [2] both use evolutionary algorithms to implement black-box patch attacks. Therefore, the authors may need further discussion.
> >
> > [1] Efficient Decision-based Black-box Patch Attacks on Video Recognition, ICCV 2023
> >
> > [2] Query-Efficient Decision-based Black-Box Patch Attack, IEEE Transactions on Information Forensics and Security

---

> > > ### Author Response · Authors · 2023-08-17
> > >
> > > We thank the reviewers for their recommended references.
> > >
> > >
> > > **[2]:** This work addresses the decision-only setting by placing localised regions of an image from the target class onto the attacked image. Under the constraint of the patch causing misclassification, the coordinates of the patch are optimised using an adapted differential evolution algorithm to minimise the size of the patch (through its l_0 norm). Whereas the imperceptibility of the patch occurs when its size becomes extremely small, we keep the size of the patch constant and reduce its visibility by approximating the region of the image it is placed upon (by minimising the l2 norm).
> > >
> > > **[1]:** This work applies a similar technique to **[2]** but for conducting patch-based adversarial attacks on video recognition networks. Specifically, they apply an adapted differential evolution algorithm to optimise the coordinate of the patch in addition to the video frame it is placed upon. Similar to [2] they construct an adversarial by placing local regions of a frame from a target class video onto the attacked video.
> > >
> > > Will will include this discussion within the final version of the manuscript, highlighting the differences with our work in terms of the addressed scenarios and methodologies.

---

### Official Review · Reviewer_Z9ie · 2023-07-07

**Soundness:** 2 fair
**Presentation:** 2 fair
**Contribution:** 3 good
**Rating:** 5
**Confidence:** 3

**Summary:**

This paper proposes a evolutionary strategy based framework for generating imperceptible adversarial patch. Compared with existing methods, the proposed method can generate invisible adversarial patch by iteratively optimizing both the patch and its position on the image.

**Strengths:**

1. The proposed method is novel and performs well on ASR and invisible.
2. The method analysis is clear and algorithm pipeline is helpful.
3. The experiments are solid and comprehensive.

**Weaknesses:**

1. Lack of the efficiency comparison with other methods.
2. Lack of the different evolutionary strategy comparison.
3. Lack of the performance on transformer based vision models.

**Questions:**

1. I am curious the computation cost of the proposed method.
2. I am curious about the performance of the proposed method with different evolutionary strategy.
3. How is the performance of the proposed method on transformer based vision models?

**Limitations:**

1. Test the computation time or memory cost of the proposed method.
2. Test the proposed method with different evolutionary strategy.
3. Test the proposed method on vision transformers.

---

> ### Author Rebuttal · Authors · 2023-08-09
>
> **W#1, Q#1, L#1:** As shown in Table 2 of our manuscript, we compare the runtime of different hyper-parameter configurations. This partially covers the evaluation of the efficiency of our proposed method. On the other hand, for the sake of fair comparisons, we fixed the computational budget to be 10,000 queries (line 208; lines 212 to 213) for all peer algorithms in our experiments. In particular, we believe the querying of the attacked deep neural network is the most computationally demanding area of an adversarial attack.
>
> **W#2, Q#2, L#2:** In this work we make use of a classic (1+1)-evolutionary strategy (ES) for patch optimization (lines 153 to 154; reference [30]). One of the most important concerns of using other ES, such as CMA-EA and PGPE with ClipUp, for patch optimization is that they usually require a large number of function evaluations to achieve optimal image approximations. This is supported in previous works in the computation creativity community (modern ES for creativity: fitting concrete $481$ images and abstract concepts; reference [44]). Unfortunately, our context is restricted to a fixed query budget. As flagged in the conclusion section, we envisage the use of surrogate model assisted approaches such as Bayesian optimization can be explored for patch optimization to further enhance the efficiency of our proposed method (lines 332 to 336).
>
> **W#3, Q#3, L#3:** At the request of the reviewer, we show the results of conducted non-targeted attacks on a Transformer based DNN classifier (Dosovitskiy et al., An Image is Worth 16x16 Words: Transformers for Image Recognition at Scale, ICLR'21 ) in **Table 1 of the global 'author rebuttal PDF'**.

---

> > ### Comment · Reviewer_Z9ie · 2023-08-16
> >
> > The anthor answers most of my questions and I will keep my rating.

---

### Official Review · Reviewer_tLpS · 2023-07-24

**Soundness:** 2 fair
**Presentation:** 3 good
**Contribution:** 2 fair
**Rating:** 5
**Confidence:** 3

**Summary:**

This paper presents a novel method for generating adversarial patches that approximates the appearance of the area it covers, achieving better or comparable performance to state-of-the-art methods while minimizing the visibility of the patch. The method uses semi-transparent, RGB-valued circles that blend into the original image with an evolutionary strategy to fine-tune each circle's properties and employs simulated annealing for optimal location selection. This paper also provides an empirical evaluation of the proposed method's effectiveness by attacking classifiers trained on the ImageNet dataset and comparing it with state-of-the-art adversarial patch methods. The contributions of this paper include a new approach to generating adversarial patches that are more difficult to detect, a comparative analysis of state-of-the-art methods, and insights into the limitations and future directions of adversarial attacks on DNNs.

**Strengths:**

1. This paper is well-written and organized, with clear explanations of the proposed method and the experimental setup. The authors provide detailed descriptions of the algorithms used and the evaluation metrics employed, making it easy for readers to understand the research.

2. The proposed CamoPatch method is a novel approach to generating adversarial patches while minimizing the visibility of the patch. The use of semi-transparent, RGB-valued circles and the evolutionary strategy is a unique contribution to the field of adversarial attacks on DNNs.

3. This paper provides a thorough empirical evaluation of the proposed method's effectiveness. The authors also provide insights into the limitations and future directions of adversarial attacks on DNNs, which adds to the quality of the research. The supplementary material is adequate as well.


**Weaknesses:**

It’s appreciated that the authors provide clear and detailed descriptions of the empirical study, though the work could be more credible if more experiments and evaluation are concerned.

1. This paper only use attack success rate (ASR) to evaluate the attack performance of the method, which is kind of limited evaluation. More metrics should be involved such as non-normalized residual (nq), original accuracy and after-attack accuracy. It would be appreciated if the experiments in terms of targeted attack are concerned.

2. When it comes to evaluating the visibility of the adversarial patches, the SSIM metric seems to be unnecessary because the difference between the proposed method and other methods in this metric is not obvious. The numerical gap is not greater than 0.02.


**Questions:**

1. As the core idea of this paper, ‘camouflaged’ can be presented in more forms such as visual comparison and evaluation on camouflage level.

2. The simulated annealing method, as the important approach to optimize the patch’s location, should be improved or replaced by a novel approach so it can be more consistent with the idea of ‘evolutionary strategy’.


**Limitations:**

It’s appreciated that the authors are up front about the limitations of their work in the paper and introduces the possible solutions in the future work. The authors may draw lessons from weakly supervised learning methods to explore adapting the proposed method to scenarios where only the predicted label of the input is available.

---

> ### Author Rebuttal · Authors · 2023-08-09
>
> **W#1:** As per requested by the reviewer, we report the original accuracy, after-attack accuracy, non-normalized residual (nq) and $l_2$ norm metrics of conducted targeted and non-targeted attacks in **Table 2 and Table 1 of the global 'author rebuttal PDF'**.
>
> **W#2:** The reason for the small numerical gap is due to the small $40 \times 40$ patch size, which is roughly 3.2% of the total number of pixels within the image (lines 207-208). Therefore, a numerical gap of 0.02 is significant, considering the small number of changed pixels.
>
> **Q#1:** To evaluate the level of camouflage, we employ SSIM and $l_2$ norm as the performance metrics in our experiments. Moreover, we provide visual comparisons of adversarial images constructed by the proposed method compared to Patch-RS (Figure 1 of our manuscript) and other state-of-the-art algorithms (Figure 3 of our manuscript). Furthermore, to facilitate a visual comparison, we provide an example of an adversarial image with the patch magnified (Figure 2 of our manuscript).
>
> **Q#2:** The selection of simulated annealing (SA) as our location optimizer is based on its proven ability to effectively explore discrete spaces and successfully avoid local optima (lines 169-170). In addition to the performance of the proposed method compared to state-of-the-art methods (Table 1 of our manuscript), we further demonstrate the superior performance achieved when using SA compared to pure random search (Table 3 of our manuscript). In future, alternative methods can be explored to improve the location optimization of a patch location, i.e. the use of surrogates (lines 327-331).
>
> **Limitations:** To enhance and advance the proposed method, we recognize the potential benefits of incorporating techniques from weakly supervised learning as suggested by the reviewer to adapt to the label-only scenario effectively. We believe there are two particularly fruitful directions of incorporating techniques from weakly supervised learning to adapt to the label-only scenario. Firstly, by using estimation techniques (lines 321-326) to label constructed adversarial images, we believe the use of weakly supervised image classification models (Hu et al., Weakly Supervised Image Classification through Noise Regularization, CVPR'19) as surrogates would improve the efficiency of the proposed method in additional to providing direction of the search for the label-only setting. Secondly, utilizing the stochastic nature of the proposed method to generate a set of non-evaluated candidate solutions, the use of weakly supervised techniques such as semi-supervised learning could be applied (Kim et al., Density Ratio Estimation-based Bayesian Optimization with Semi-Supervised Learning, Corr'23) to train a surrogate model on both evaluated and non-evaluated adversarial images. Thereby using the surrogate to select predicted-optimal solutions for evaluation.

---

### Author Rebuttal · Authors · 2023-08-09

Attached is the manuscript containing the tables and figures of reviewer requested experiments and plots.

---

### Decision · Program_Chairs · 2023-09-21

**Decision:**

Accept (poster)

**Comment:**

The paper proposes a localized adversarial attack approach that creates camouflaged perturbed patches using an evolutionary approach. Initial concerns have been well addressed in the author's response, which leads to all 5 reviewers voting for acceptance. AC concurs and believes the paper will be a valuable result for the community. Authors are strongly suggested to reflect the promised changes in the final version and add clarification where requested.